# Synthesis of Niobium-Alumina Composite Aggregates and Their Application in Coarse-Grained Refractory Ceramic-Metal Castables

**DOI:** 10.3390/ma14216453

**Published:** 2021-10-27

**Authors:** Tilo Zienert, Dirk Endler, Jana Hubálková, Gökhan Günay, Anja Weidner, Horst Biermann, Bastian Kraft, Susanne Wagner, Christos Georgios Aneziris

**Affiliations:** 1TU Bergakademie Freiberg, Institute of Ceramics, Refractories and Composite Materials, Agricolastr. 17, 09599 Freiberg, Germany; dirk.endler@ikfvw.tu-freiberg.de (D.E.); Jana.Hubalkova@ikfvw.tu-freiberg.de (J.H.); aneziris@ikfvw.tu-freiberg.de (C.G.A.); 2TU Bergakademie Freiberg, Institute of Materials Engineering, Gustav-Zeuner-Str. 5, 09599 Freiberg, Germany; goekhan.guenay@iwt.tu-freiberg.de (G.G.); weidner@ww.tu-freiberg.de (A.W.); biermann@ww.tu-freiberg.de (H.B.); 3Karlsruhe Institute of Technology, Institute for Applied Materials-Ceramic Materials and Technology, Haid-und-Neu Straße 7, 76131 Karlsruhe, Germany; bastian.kraft@kit.edu (B.K.); susanne.wagner@kit.edu (S.W.)

**Keywords:** refractory composite, aggregate synthesis, castable

## Abstract

Niobium-alumina aggregate fractions with particle sizes up to 3150 µm were produced by crushing pre-synthesised fine-grained composites. Phase separation with niobium enrichment in the aggregate class 45–500 µm was revealed by XRD/Rietveld analysis. To reduce the amount of carbon-based impurities, no organic additives were used for the castable mixtures, which resulted in water demands of approximately 27 vol.% for the fine- and coarse-grained castables. As a consequence, open porosities of 18% and 30% were determined for the fine- and coarse-grained composites, respectively. Due to increased porosity, the modulus of rupture at room temperature decreased from 52 MPa for the fine-grained composite to 11 MPa for the coarse-grained one. However, even the compressive yield strength decreased from 49 MPa to 18 MPa at 1300 °C for the fine-grained to the coarse-grained composite, the latter showed still plasticity with a strain up to 5%. The electrical conductivity of fine-grained composite samples was in the range between 40 and 60 ^S^/cm, which is fifteen magnitudes above the values of pure corundum.

## 1. Introduction

Refractory ceramics are typically used in metallurgy applications. They are characterised by good mechanical properties, good thermal shock behaviour as well as producibility of large, coarse-grained components by water-based casting processes, such as pressure slip casting [1] or castable technology [2].

All physical and mechanical properties of refractory castables are depending on the firing temperature. The higher the firing temperature of the castables the more pronounced is the ceramic bonding of the fine matrix. Hence, the high degree of sintering results in lowering porosity and enhancing both modulus of rupture as well as Young´s modulus [3]. The open porosity of conventional alumina castables bonded with high alumina cement fired at 1000 °C varies from 20% to 30% depending on the type of aggregates and cement used [4]

After firing at 1600 °C, the low cement castables (LCC) and cement-free castables (NCC) based on alumina have an open porosity in the range from 15 to 20% [5]. The Young´s moduli of alumina at ambient temperature of such castables range between 60 GPa and 100 GPa depending on the chemical and granulometric composition [6]. For the industrial application of refractory castables, a detailed evaluation of predominant stress conditions is essential. If a high erosion resistance is required, the microstructure of the castables should be highly sintered with a low degree of porosity. Nevertheless, the high degree of sintering involves a pure brittle behaviour accompanied by a high susceptibility to cracking and subsequent spalling. Otherwise, if high thermal shock resistance is of prime importance, a low Young´s modulus and relative high porosity are beneficial, provided that the mechanical stability, i.e., sufficient modulus of rupture, is ensured. The shrinkage of coarse-grained castable during the sintering should not exceed a value of 2% to prevent crack formation.

Ceramic-matrix and metal-matrix composites have been investigated since several decades [7]. In particular, the combination of ceramics with refractory metals is of interest for high-temperature and wear applications, e.g., corundum-molybdenum [8,9,10,11]; mullite-molybdenum [12,13,14]; zirconia-(tantalum, niobium) [15,16]. In that works, fine-grained powders were used to produce dense sintered composites focusing on improving the mechanical properties of the ceramic part by the combination of the ceramic’s high strength and the metal’s toughness and grain fining ability [17]. High shrinkage on sintering, limited thermal shock ability and/or the sintering technology as hot isostatic pressing are restricting the maximum size of fine-grained components in fabrication.

The concept of coarse-grained refractory composites based on tantalum-alumina and niobium-alumina was recently introduced [18] combining the fields of powder metallurgy and castable technology. Shrinkage values of ≈1% and good mechanical properties at ambient temperature as well as in the temperature range of 1300–1500 °C [19] were obtained. Low shrinkage on sintering causes also low values of residual stresses in the refractory ceramic, which enables one to fabricate large refractory components. Refractory composite castables based on coarse-grained tabular alumina and fine-grained metal of 11 vol.% and 21 vol.% substituing the alumina matrix showed the best results regarding physical and mechanical properties. It was suggested to produce pre-synthesised composite aggregates for further composite castables.

The approach of refractory ceramic-metal castables aims to leverage the synergy effects of both refractory ceramics and refractory metals and hence to develop smart key components for metallurgical industry with targeted functionalisation, such as integrated electrodes for non-wetting behaviour [20], ladle sliding gate plates with enhanced thermal shock properties, special casting nozzles with anti-clogging properties or integrated heating elements.

The wetting behaviour of a liquid metal/ceramic system depends on the bonding characteristics of both system components as well as on the magnitude of interactive forces at the interface. It is characterised by the contact angle of the liquid drop formed on the solid substrate indicating the chemical affinity between the two materials and by the work of adhesion as a measure of interface adhesion strength [21]. Speaking about industrial applications of refractory ceramics, it is necessary to know the interfacial properties (surface/interfacial tension and surface segregation) of melts and characteristics of solids (surface roughness, impurities, porosity) including the operating conditions such as temperature, time, gas composition and pressure [22,23].

The aim of the present work is to investigate the feasibility to produce refractory niobium-alumina composite material in a novel two-step castable procedure. The first step comprises the synthesis of targeted niobium-alumina composite aggregrates while the second step involves the production of coarse-grained refractory castables utilising the pre-synthesised composite aggregates. Such a fractal design is needed in order to ensure a coherent composition at different aggregate size scales and hence to achieve an electrical conductivity and proper mechanical properties (thermal shock resistance, sufficient high-temperature strength). For this purpose, the metallic phase should constitute 65 vol.% of the composite material.

## 2. Methods

### 2.1. Sample Preparation

Initially, the preliminary fine-grained material had to be synthesised and subsequently crushed and sieved in fractions to obtain refractory metal-ceramic aggregates of different sizes. The composite niobium-alumina aggregates were then used to produce coarse-grained refractory composite castables.

The base material for the production of aggregates with an intended niobium amount of 65 vol.% was synthesised from raw materials (particle sizes and chemical information are listed in Table 1) as follows. Powders of niobium (EWG Wagner, Weissach, Germany), alumina (Martoxid, Martinswerke, Germany), a hydratable alumina binder (Alphabond 300, Almatis, Ludwigshafen, Germany) and dispersing alumina (ADS-1, Almatis, Ludwigshafen, Germany) were first dry mixed for one minute using an Eirich mixer (Gustav-Eirich Maschinenfabrik, Hardheim, Germany) followed by wet mixing for four minutes with step-wise water addition until almost self-flowability was achieved. The composition of the fine-grained composite material is given in Table 2. This mixture was then filled in steel molds with dimensions of 25mm×25mm×150mm by vibrational assisted casting. After setting at room temperature for 48 h, the samples were demolded and dried for 24 h at 130 °C in air atmosphere. Afterwards, the prisms were sintered at a temperature of 1600 °C for 4 h in a corundum tube furnace under flowing argon atmosphere, which was purified by a titanium getter. This material is hereinafter referred to as *fine-grained composite*.

In a further step, the synthesised fine-grained prisms were broken and crushed using a jaw crusher (BB50, Retsch, Haan, Germany) with jaws made from hard metal (92% WC–8% Co). The crushed material was sieved into the four aggregate classes 0–45 µm, 45–500 µm, 500–1000 µm and 1000–3150 µm.

The particle size distribution (Bettersizer S3 plus, 3P Instruments GmbH & Co. KG, Odelzhausen, Germany), density and phase assemblage using X-ray powder diffraction (XRD) (Empyrean, Malvern Panalytical GmbH, Kassel, Germany) in combination with Rietveld refinement using HighScore Plus 4.8 [24] of the synthesised composite aggregates were determined. XRD measurements were done using Cu-K${}_{{}_{1}}$ radiation between 15–140° 2Θ with 0.0143° step size and an exposion time of 160 ^S^/step.

Densities were measured using mercury intrusion porosimetry (AutoPore V 9600, Micromeritics Instrument Corp., Norcross, GA, USA) on particles according to DIN ISO 15901-1:2019-03 and gas pycnometry with helium (AccuPyc 1340TEC, Micromeritics Instrument Corp., Norcross, GA, USA) on fine-powdered material according to DIN 66137-2:2019-03 of each aggregate class. In addition, true densities of each aggregate class were calculated based on the results of the XRD/Rietveld analysis. Particle morphology of the synthesised aggregates was investigated based on laser-scanning microscopy (VK-X1000, Keyence Deutschland GmbH, Neu-Isenburg, Germany).

### 2.2. Design of Coarse-Grained Castables

Particle size distributions of castables can be mathematically described using particle packing models expressing the cumulative sum curve or *cumulative percent finer than particle size d* CPFT(*d*). A commonly used model is the Dinger–Funk model [25], which allows to take the minimum and maximum particle sizes ($d_{\min}$ and $d_{\max}$) into account. Recently, this model was modified by Fruhstorfer [26] to
(1)$$CPFT_{\mathrm{mod}-\mathrm{DF}}\left(d\right)=100\%\cdot \frac{d^{n(d)}-d_{\min}^{n(d)}}{d_{\max}^{n(d)}-d_{\min}^{n(d)}}$$
using the particle-size dependent distribution modulus
(2)$$n\left(d\right)=n_{\min}+d\cdot \frac{n_{\max}-n_{\min}}{d_{\max}}\;,$$
where $n_{\min}$ and $n_{\max}$ are the minimum and the maximum distribution modulus, respectively. The flowability of a castable can be related to its particle distribution and hence, $n_{\min}$ and $n_{\max}$ can be used as parameters to describe a chosen material mixture. In case of tabular alumina with a maximum particle size of 3150 µm, best properties in terms of flowability, density and pore sizes of the castable were found for $n_{\min}=0.28$ and $n_{\max}=0.8$ [27].

The synthesised niobium-alumina aggregates were wet mixed with the hydratable alumina binder Alphabond 300 and the reactive alumina CL370 (Almatis, Ludwigshafen, Germany) according to the recipe as discussed in Section 3.3 using a concrete laboratory mixer (ToniMAX, Toni Baustoffprüfsysteme GmbH, Berlin, Germany). The coarse-grained mixture was then casted into the prismatic molds, set, dried and sintered following the same procedure as the preliminary, fine-grained material. The sintered castable samples will be referred to as *coarse-grained composite* hereinafter.

### 2.3. General Sample Characterisation

The fine-grained as well as the coarse-grained composites (four prisms each) were characterised in terms of shrinkage, envelope density and open porosity according to DIN EN 993-1:2019-03; elastic constants according to DIN EN 843-2:2006 by the ultrasonic procedure (UKS-D device, GEOTRON-ELEKTRONIK, Pirna, Germany).

The cold modulus of rupture (MOR) of the fine-grained composite (two prisms) and the coarse-grained composites (three prisms) was determined using an universal testing machine TIRAtest 28100 (TIRA GmbH, Schalkau, Germany). In order to achieve a uniform distribution of bending moment, four point bending setup with a support span of 125 mm, a load span of 62.5 mm, a pre-load of 20 N and a loading-rate of 150 ^N^/s was applied.

On cut slices, the interior microstructure and phase assemblage of the prisms were studied by scanning electron microscopy (SEM) using back-scattered electron (BSE) and secondary electron (SE) contrast together with energy dispersive X-ray spectroscopy (EDS) (Philips XL30 ESEM FEG, Amsterdam, the Netherlands). Some cut slices were also crushed and powdered using an agate mortar and analysed by XRD/Rietveld refinement.

#### 2.3.1. CT Measurements

The macrostructure of the composite material was analysed using a microfocus X-ray tomograph CT-ALPHA (ProCon X-ray GmbH, Sarstedt, Germany) equipped with a transmission X-ray tube FXE-160.20/25 (Feinfocus, Garbsen, Germany) and a flat panel X-ray detector Dexela 1512 (Perkin Elmer, Rodgau, Germany).

The µ-CT was operated at 150 kV and 120 µA using a 0.6 mm copper filter. The exposure time was set to 2 s. The volume data were reconstructed by means of the software Volex 6.0 (Fraunhofer EZRT, Fürth, Germany) with a voxel size of 9.8 µm. The reconstructed volume data were visualised using the software VG Studio MAX 2.2 (Volume Graphics, Heidelberg, Germany) and quantified with the software MAVI 1.5.3 (Fraunhofer ITWM, Kaiserslautern, Germany). The image processing comprised a cropping step in order to cut out a defined volume of the reconstructed data (350×350×1050voxels) followed by a segmentation step in order to transform the grey scale into binary image. The binarisation was performed using Otsu´s method [28] being based on a global thresholding strategy. Subsequently, the field features (volume density and total porosity) were determined.

#### 2.3.2. High-Temperature Compressive Strength Measurements

Quasi-static compression tests were performed on both, fine-grained and coarse-grained composites (one specimen each). The cylindrical specimens with 12 mm in diameter and 20 mm in height were drilled from sintered prisms. The compression tests were conducted at an initial strain rate of $7.5\times 10^{-4}1\;/\;s$ at 1300 °C up to a possible maximum strain of 30% using an electro-mechanical, high-temperature testing machine (Z020, Zwick Roell, Haan, Germany) with a protective argon gas chamber (Maytec, Olching, Germany) integrated into the testing machine. To prevent the oxidation of the specimens, the test chamber was evacuated to 0.8 mbar vacuum and then filled with argon twice. Thus, the tests were performed under argon atmosphere and ambient pressure. The presence of oxygen was controlled by an oxygen sensor (Stange Elektronik, Gummersbach, Germany).

Before the tests, the lower piston was moved upwards with a speed of 0.1 mm/min to apply a pre-load of 5 N to the specimens. The heating of the specimens was carried out inductively via a medium-frequency induction generator HF 5010 (TRUMPF Hüttinger, Ditzingen, Germany) with a heating rate of 30 ^K^/*s* and a water-cooled copper coil. The temperature was detected by a pyrometer Metis MS09 (Sensortherm GmbH, Steinbach/Ts, Germany) with a wavelength of 0.9 µm and an emission coefficient of 0.93. In case of the coarse-grained composite, a metallic susceptor made of Mo-alloy TZM was used, since the material could not be heated by induction. The specimens were kept 20 min under the pre-load after reaching the test temperature to achieve a homogeneous temperature field over the entire height of the specimens. More details of the test procedure were given elsewhere [19].

#### 2.3.3. Electrical Conductivity Measurements

Two samples for electrical conductivity measurements were prepared as follows. First a powder mixture of 65 vol.% niobium and 35 vol.% alumina (calcinated alumina CT9FG, Almatis, Ludwigshafen, Germany) was dry mixed and homogenised, pressed to cylinders and finally sintered under argon atmosphere at a temperature of 1600 °C for 4 h.

To remove the formed sinter skins and to create a plane surface, the circular faces of the samples were grinded using SiC paper (P400 and P600). Then the surfaces were sputter-coated with gold, using a Quorum Q150T ES (Quantum Design GmbH, Darmstadt, Germany). Due to surface roughness, the circular faces were coated with a layer of silver paste to ensure that the whole surface area of the sample was connected by one continuous electrode.

The investigations were performed using a four-point-measurement setup, which is schematically shown in Figure 1. During the tests, currents of 1 mA, 10 mA and 100 mA were applied to the samples working with a Keithley 220 Programmable Current Source (Keithley Instruments, Cleveland, OH, USA) and the resulting voltages were measured using a Keithley 2000 Multimeter (Keithley Instruments, Cleveland, OH, USA). The resistance *R* of the samples was calculated with
(3)$$R=\frac{U}{I}\;,$$
where *I* is the applied current and *U* is the resulting voltage [29]. The specific resistance ρ was calculated according to
(4)$$\rho =\frac{R\cdot A}{l}$$
where *A* is the surface area of the prepared electrode and *l* the distance between the prepared sample surfaces.

## 3. Results and Discussion

### 3.1. Fine-Grained Composite

SEM micrographs of the sintered fine-grained composite can be seen in Figure 2. The niobium particles are homogeneously distributed within a fine-grained matrix of alumina (Figure 2a). Pores with a diameter up to ≈100µm were only randomly distributed within the sample volume and, therefore, less visible in the presented micrographs. The impurity phase NbO identified by EDS measurements was frequently present within niobium particles as it is shown in Figure 2b. Ternary oxide formation, e.g., of AlNbO${}_{4}$ as reported in [19], at the interface alumina/niobium oxide was not detected by SEM/EDS. Furthermore, the niobium material showed only a less pronounced tendency for surface diffusion along the alumina grains in comparison to the previous case of a mixture with a tabular alumina material [18].

### 3.2. Synthesised Composite Aggregates

#### 3.2.1. Morphology and Particle Size Distributions

Photographs of the aggregate fractions 500–1000 µm and 1000–3150 µm are presented in Figure 3. The crushing process produced mainly plate-like shaped particles. The particle height is much smaller than the length and width. Quantitatively spoken, the ratio of height to the maximal length is approximately 0.5–0.7. Within this overall morphology, two types can be distinguished. First, roughly two third of the particles approximate an imaginary plate where a relatively flat plateau in height-direction was formed. For the other one third of particles, a more or less pronounced maximum peak can be observed. These two shape categories are exemplary illustrated in Figure 4 determined on particles from the 1000–3150 µm aggregate fraction.

The characteristic particle sizes $d_{10}$, $d_{50}$ and $d_{90}$ of the synthesised composite aggregates measureable with laser granulometry are listed in Table 3. A continuous uniform particle size distribution was assumed for the aggregate class 1000–3150 µm.

#### 3.2.2. Density and Porosity

The envelope density of the fine-grained composite was determined with the Archimedes principle to 5.66 g/$cm^{3}$ including 18% open porosity. A skeleton density of 6.73 g/$cm^{3}$ was determined with mercury intrusion porosimetry (MIP) resulting in an envelope (bulk) density of 5.78 g/$cm^{3}$ and hence an (open) porosity of 16.5% as presented in Table 4. The total porosity of the fine-grained composite was estimated based on the CT image as shown in Figure 5 to 32.60%, which gives an estimated closed porosity in the range of 10–15%. However, the estimated porosity must be interpreted with caution. There are several constraints affecting the quantitative CT analysis. Due to the limited resolution, e.g., reconstructed voxel size, pores smaller than approximately 20 µm were unevitably neglected. Furthermore, uneven thickness of the scanned sample resulted in an irregular grey value distribution along the width of the sample, as it can be seen in Figure 5. Therefore, the image processing, particulary the binarisation step, is a subject of errors.

#### 3.2.3. XRD Analysis

Detailed results of the Rietveld refinement of the aggregate fractions are listed in Table 5. The crushing procedure of the sintered fine-grained composite induced phase separation resulting in alumina enrichment in the finest aggregate fraction 0–45 µm and niobium enrichment in the aggregate fraction of 45–500 µm. For the aggregate fractions larger than 500 µm, the volume ratio of niobium to corundum is around 55:40, whereas for the aggregate fractions 0–45 µm and 45–500 µm this ratio is 40:57 and 60:35, respectively. Phase separation due to crushing is also confirmed by the results of the density measurements using helium pycnometry as given in Table 4. The largest density was determined for the 45–500 µm aggregate fraction, which is in good agreement with the calculated true densities based on the XRD results. However, the true densities determined by helium pycnometry are 2.5–4% larger than the calculated ones, which reflects the error of the quantitative phase analysis.

In all fractions, the impurity phases -Nb${}_{2}$C [30] (adopted from #ICSD 33575) and NbO [31] (#ICSD 27574) were detected each with an amount between ≈1–2 vol.%. It was found that the volume content $V_{\mathrm{impurity}}$ of NbO and -Nb${}_{2}$C in each aggregate fraction is linearly depending on the corresponding niobium content $V_{\mathrm{Nb}}$, which can be described by
(5)$$V_{\mathrm{impurity}}=f_{\mathrm{impurity}}\cdot V_{\mathrm{Nb}}\;.$$

For NbO and -Nb${}_{2}$C, the factors $f_{\mathrm{NbO}}$ and $f_{-\mathrm{Nb}}{}_{2}C$ were obtained by least-square fitting to 0.0368 and 0.0369, respectively, resulting in $f_{\mathrm{NbO}+-\mathrm{Nb}}{}_{2}C$ of 0.0737. Such linear dependency means that the grains of NbO and -Nb${}_{2}$C are always connected to the niobium crystals. Crushing did not result in a separation of the formed Nb-based impurities from the niobium raw material.

As an example, the result of the XRD measurement and the Rietveld refinement of the aggregate fraction 1000–3150 µm can be seen in Figure 6a. The niobium-related reflections were described using two niobium phases each with a different UVW profile and lattice parameter. For the 1000–3150 µm aggregate fraction, lattice parameter of a=3.31377Å and a=3.30858Å were obtained for the phases marked as ’Niobium-1’ and ’Niobium-2’ in Figure 6, respectively. Using two niobium phases during Rietveld refinement led to a large drop in the resulting weighted profile R-values $R_{\mathrm{wp}}$ in comparison to using only one phase. For a detailed view, the refinement result of the niobium (132) reflection is shown in Figure 6b. The calculated profiles of the two niobium phases are different, which can be related to a difference in crystallite size. Our material has at least two different morphologies of metal grains. One are the particles of the raw material, which are connected to NbO particles as shown in Figure 2b. The second type of metal grains are the ones that originate from diffusion along the surface of the corundum particles (compare also with the reported microstructures in [18]). The difference in the lattice parameter of the two niobium phases can be related to a different chemistry. It can be assumed that the niobium diffused along the corundum grains has a larger amount of dissolved impurity elements (such as O, K, Na) in comparison to the relatively large particles of the niobium raw material. However, it should be mentioned that no differences in chemistry were observed by SEM/EDS measurements. Most likely such differences are below the detection limit of the EDS method. The spread of lattice parameters could be also explained by the presence of residual stress in niobium caused by the crushing process.

Figure 7 shows the determined lattice parameters of the phases Niobium-1 and Niobium-2 for all synthesised aggregate fractions. The dependence of the lattice parameter of Niobium-1 on the particle size shows a pronounced maximum for the aggregate fraction 45–500 µm. Contrarily, the lattice parameter of Niobium-2 reveal an opposite tendency with a minimum for the aggregate fraction 45–500 µm. It can be assumed that the chemically inhomogeneous niobium entails local differences in hardness and toughness leading to the observed phase separation during crushing.

### 3.3. Coarse-Grained Castable Recipe

Using $n_{\min}=0.28$ and $n_{\max}=0.8$ in Equation (Equation 1) as mentioned in Section 2.2, a volume ratio of the aggregate classes 0–45 µm:45–500 µm:500–1000 µm:1000–3150 µm of 27.6:23.2:9.05:40.1, which is 0.69:0.58:0.23:1 normalised to the largest aggregate class, should be utilised in the castable mixture.

The amount of raw material was a strongly limiting factor for obtaining the intended volume ratios of the synthesised aggregate fractions. Nevertheless, the actual yields of the single aggregate fractions during the crushing procedure did not satisfy the requirements on the adequate volume amounts. Therefore, it was necessary to adjust the volume ratio of the aggregate classes. Finally, a mass ratio of 5.7:33.6:28.2:32.5 of the aggregate classes 0–45 µm:45–500 µm:500–1000 µm:1000–3150 µm was approved for producing the coarse-grained castable. Using the above discussed densities of the synthesised aggregate fractions, a normalised volume ratio of 0.18:1.03:0.87:1 of the aggregate fractions was realised.

For the castable mixture, approximately 15 vol.% of alumina-based fines and binder were additionally added to the synthesised aggregates. The castable component’s particle size distributions are shown in Figure 8 according to the derived recipe (see Table 6). Finally, the normalised volume ratio of the castable mixture was 0.95:1.23:0.43:1, resulting in an increased amount of particles of the aggregate classes 45–500 µm and 500–1000 µm in comparison to the optimal values shown above. The particle distribution of the coarse-grained composite is characterised by Equation (Equation 1) using $d_{\min}=0.01\;µm$, $d_{\max}=3150\;µm$ with $n_{\min}=0.2716\left(30\right)$ and $n_{\max}=0.295\left(24\right)$.

### 3.4. Coarse-Grained Composite

SEM micrographs of the sintered coarse-grained composite are shown in Figure 9 and Figure 10. In comparison to the sintered fine-grained composites (Figure 2), the niobium particles are inhomogenuously distributed as the niobium-rich pre-sintered coarse-grained particles are now embedded within an alumina-rich matrix based on CL370 and the de-hydrated Alphabond300 binder as it can be seen in Figure 9. In addition, large pores with a few hundred micrometers in diameter were observed regularly within the microstructure (Figure 9a). In Figure 9b, the indicated particle boundaries of the pre-synthesised aggregates are drawn for more clarity. However, the phase assemblage of the niobium particles did not change during the second sintering process in comparison to the sintered fine-grained composite. Still, niobium oxide is present within some of the niobium particles without any SEM/EDS-detectable chemical reaction with alumina (see Figure 10).

The envelope density of the coarse-grained composites was determined to 4.57 g/$cm^{3}$ with 30.4% open porosity, whereas MIP measurements on fragments of the coarse-grained prisms give an envelope (bulk) density of 5.17 g/$cm^{3}$ including 25.6% (open) porosity. These differences can indicate surface effects. However, both measurements point up that the coarse-grained composite exhibit a lower density than the fine-grained one (see Table 4).

### 3.5. Comparison of Mechanical Properties and Elastic Constants of the Fine- and Coarse-Grained Composites

The determined shrinkage of the sintered fine-grained composite is 4.27 ± 0.42% whereas the coarse-grained composite evinces a significant reduced shrinkage values of 1.61 ± 0.13%.

The determined values of Young’s modulus *E*, shear modulus *G* and Poisson’s ratio of the fine-grained and the coarse-grained composites are listed in Table 7. The Young’s modulus of the coarse-grained composite is only ≈50% of the fine-grained one, whereas the shear modulus and Poisson’s ratio dropped by roughly one third. It can be assumed that the relatively low elastic constants originate from the high porosity of the coarse-grained composite. As it is shown in Table 4, the determined open porosity increased from 18 to 20% in case of the fine-grained composite to 25–30% for the coarse-grained samples.

The four-point bending strength also decreased from 52 MPa to 11 MPa comparing the fine- and coarse-grained composites (see Table 7). The fine-grained composite has a similar strength as a formerly produced fine-grained niobium-alumina composite containing 11 vol.% metal [18].

### 3.6. High-Temperature Compressive Strength

Figure 11a shows the high-temperature compressive behaviour of fine-grained and coarse-grained composite materials at 1300 °C. The fine-grained composite has a significantly higher strength compared to coarse-grained composite with a compressive yield strength of 49 MPa, while the coarse-grained specimen achieved 18 MPa only. In addition, pronounced plastic deformation occurred in the fine-grained composite, as it is visible from Figure 11b. After reaching 10% of strain, the stress on the fine-grained composite slightly increased due to strain hardening within the material. Furthermore, the effective cross-section of the sample was increased. The specimen was then compressed well up to 30% of strain. On the other hand, the coarse-grained composite showed more brittle behaviour with limited plasticity and failed after approximately 4–5% of strain.

Figure 11b compares the appearence of the specimens before and after the tests. As it can be seen, some cracks occurred on the fine-grained specimen during the test, however, in spite of these cracks, the specimen showed good plastic behaviour and did not fail. The coarse-grained composite showed fracture to larger fragments as it is clearly visible in Figure 11b. Thus, the specimen showed limited plasticity and more brittle behaviour and the specimen failed. The presence of higher porosity is considered to be the fundamental reason for the brittle behaviour.

### 3.7. Electrical Conductivity

Two samples were produced with a density of 5.3 g/$cm^{3}$ and 5.2 g/$cm^{3}$, and a determined electrical conductivity of 40.58 ^S^/cm and 59.14 ^S^/cm, respectively. The high electrical conductivity is caused by the formation of a conducting network of the metallic component through the composite material. According to [32], the specific electrical resistance of pure niobium is in the range of $1.46\cdot 10^{-5}-1.70\cdot 10^{-5}\;\cdot cm$, resulting in an electrical conductivity in the range of 68.49–58.82kS/cm. A comparison of these values with the results of the investigated niobium-alumina composites shows a decrease of the electrical conductivity in the order of three magnitudes.

The lower electrical conductivity is caused by the composition of the material, made from 65 vol.% electrically conductive niobium and 35 vol.% isolating alumina. Furthermore, the sample density has to be considered, since the samples exhibited a geometric density of about 75 % theoretical density. The influence of porosity on electrical conductivity has been studied elsewhere and is reported to decrease with increasing porosity [33,34,35].

## 4. Conclusions

Niobium-alumina aggregates were successfully synthesised and their application in coarse-grained refractory castables was demonstrated. It was found that the crushing process of the fine-grained 65/35 vol.% niobium-alumina composite induced phase separation resulting in niobium contents between 40 and 60 vol.%, depending on the aggregate class. The porosity of the fine-grained composite might play a role here as well as the amount of diffusion of the niobium along the grain boundaries of the alumina material. Such interactions must be understood in more detail for a better designing of refractory composite aggregates and the related coarse-grained castables.

Obviously, the physical properties and composition of the aggregates are depending on the aggregate fractions, i.e., on the aggregate size. In addition, the particle morphology and the porosity of the aggregates play an important role in packing density, and water demand for self-flowability of the castable mixture. It was shown that the particle morphology variety can not be ignored. Therefore, a systematic analysis of particle morphologies should be part of further works.

A relatively high water content of almost 30 vol.% was necessary for achieving a vibration-assisted flowability of the fine- and coarse-grained castables. Therefore, the porosity of the coarse-grained composite increased due to aggregate properties (roughness, relatively high open porosity) in comparison to the fine-grained composite, which is also reflected by lower values of elastic constants and mechanical strength. However, a strain value of almost 5% was achieved for the coarse-grained composite in compression at a temperature of 1300 °C demonstrating good thermal shock properties in refractory ceramic applications.

For the coarse-grained composite, shrinkage of 1.6% was determined after sintering. In combination with the high metal content, the material can be used to produce large components without the risk of shrinkage cracks on sintering.

The electrical conductivity of the refractory composite strongly depends on chemistry and porosity. The investigated pressed and sintered fine-grained composites had a similar porosity as the casted ones. Thus, the values of electrical conductivity should be in the same order of magnitude for the two materials. With 40–60 ^S^/cm, the electrical conductivity of our fine-grained composite is only one magnitude below the one of silicon carbide [36] but fifteen magnitudes above the electrical conductivity of pure alumina [37]. Thus, the (coarse-grained) material is a candidate for high-temperature heating elements or for refractory linings with voltage-controlled metal melt wetting. For such a purpose, the composite must be used under protective atmosphere or the surface must be coated, e.g., with alumina, for oxidation resistance. In further works, the electrical conductivity of the refractory composite should be investigated on densely sintered samples to study the influence of chemistry as well as on samples containing different levels of closed and open porosity.

Mechanical and functional properties of coarse-grained refractory composite are mainly depending on the properties of the used aggregates. Despite the presented splintery-shaped particles produced by crushing, circular-shaped, porous aggregates based on alginate gelation [38] or almost dense particles based on pelletising can improve thermo-mechanical properties of coarse-grained refractory composites. It is a challenging task to find the best-fitting material solution for high temperature application simultaneously offering a good thermal-shock behaviour and sufficient mechanical properties [39]. By assembling of pre-synthesised dense and porous aggregates in different size-dependent volume ratios in a castable offset could be a promising way to achieve this objective, particularly for large refractory components.

## Figures and Tables

**Figure 1 materials-14-06453-f001:**
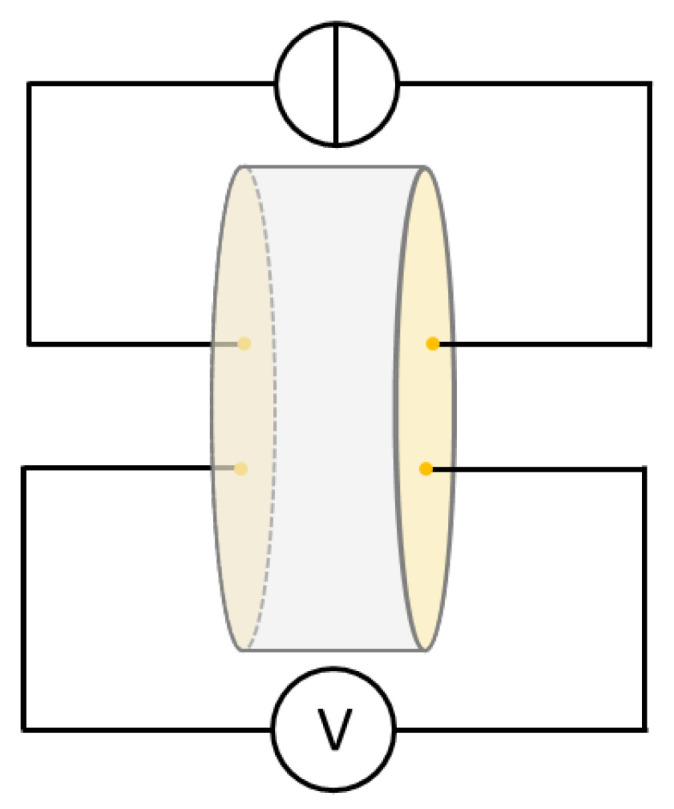
Schematic illustration of electrical conductivity measurement setup.

**Figure 2 materials-14-06453-f002:**
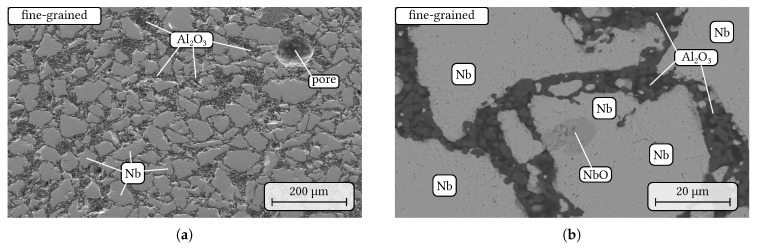
SEM micrographs of the fine-grained composite (**a**) SE contrast, overview and in detail (**b**) BSE contrast, NbO impurity within Nb grains surrounded by a finer grained alumina matrix.

**Figure 3 materials-14-06453-f003:**
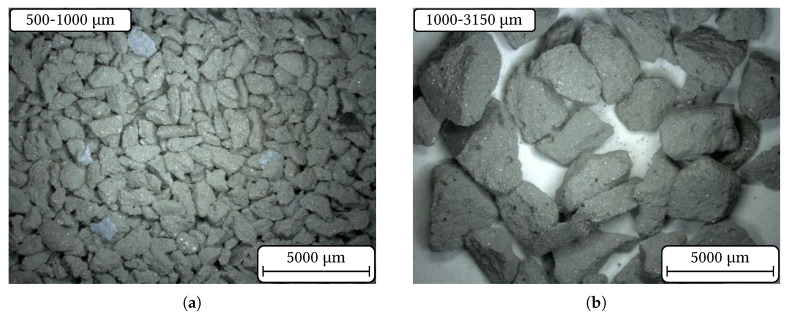
Photographs of the synthesised aggregate fractions (**a**) 500–1000 µm and (**b**) 1000–3150 µm.

**Figure 4 materials-14-06453-f004:**
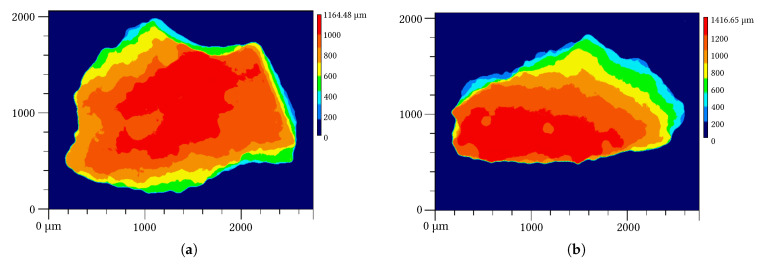
Height maps of particles from the aggregate fraction 1000–3150 µm determined with laser-scanning microscopy showing two types of plate-like particle morphologies: (**a**) relatively flat surface plane and (**b**) a more or less pronounced maximum in the vertical direction.

**Figure 5 materials-14-06453-f005:**
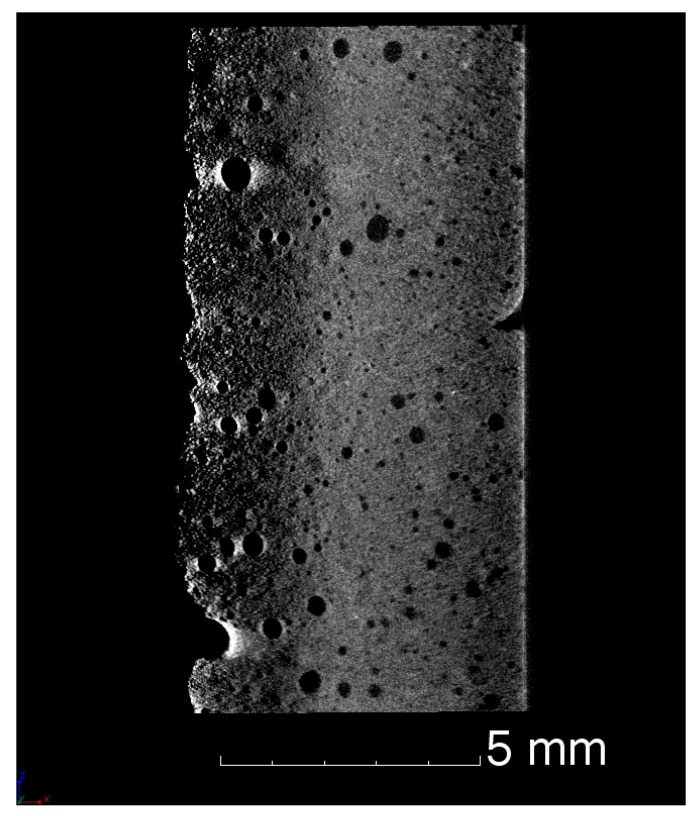
CT image of the fine-grained composite showing inhomogeneously distributed pores with a diameter larger than 100 µm.

**Figure 6 materials-14-06453-f006:**
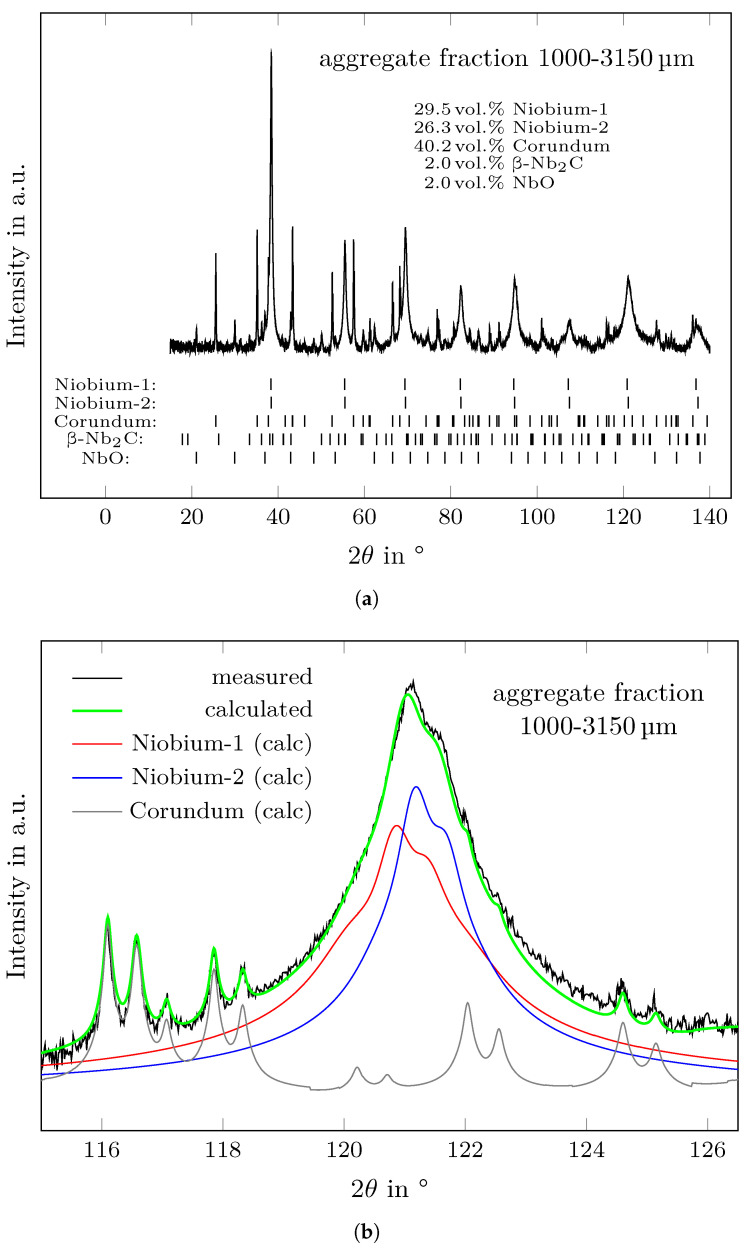
Results of XRD measurement and Rietveld refinement of the 1000–3150 µm aggregate fraction. (**a**) Determined diffraction pattern and calculated positions of each phase reflections between 15 and 140° 2θ and (**b**) enlarged view showing the calculated intensities of the niobium (132) reflection.

**Figure 7 materials-14-06453-f007:**
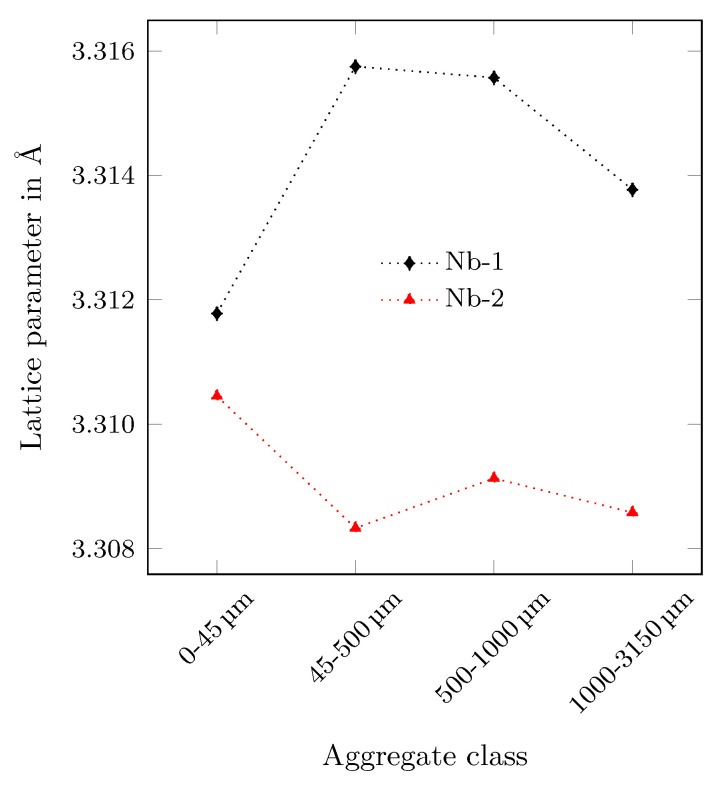
Lattice parameters of the two niobium phases used for Rietveld refinement obtained for all synthesised aggregate fractions.

**Figure 8 materials-14-06453-f008:**
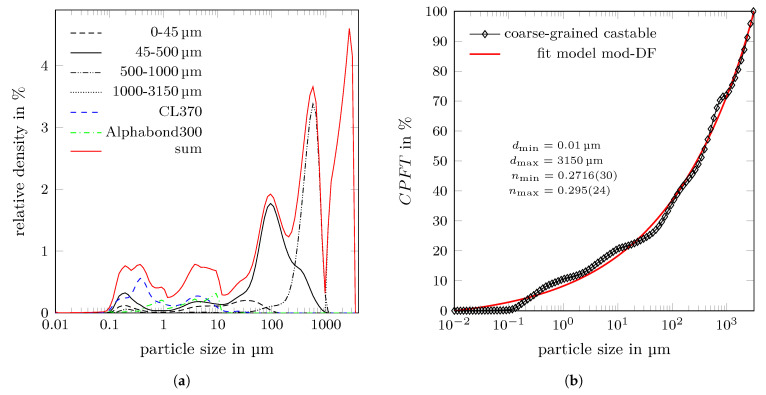
(**a**) Determined particle size distributions of the used castable components as they contribute to the castable’s particle size distribution (calculated sum distribution); (**b**) CPFT of the particles of the coarse-grained castable together with the results of the fit using the modified Dinger-Funk distribution model (see Equation (Equation 1)).

**Figure 9 materials-14-06453-f009:**
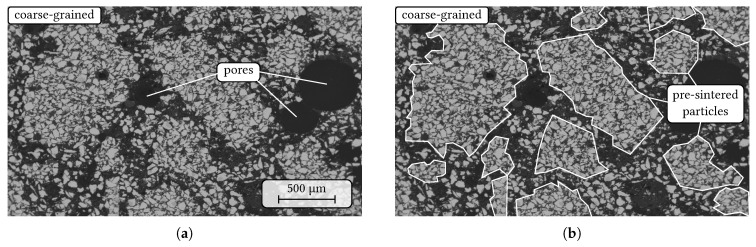
SEM micrograph (BSE contrast) of the coarse-grained composite with niobium (light grey) and alumina (dark grey). (**a**) overview with pores up to a few hundreds micrometers size, (**b**) the same micrograph with indicated particle boundaries of the pre-sintered aggregates.

**Figure 10 materials-14-06453-f010:**
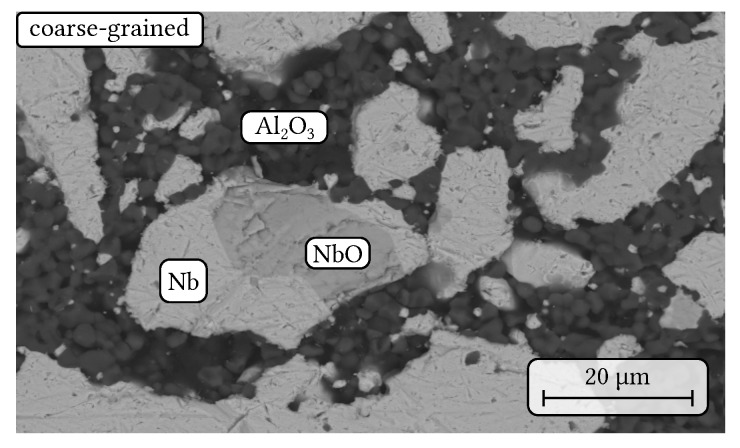
SEM micrograph in BSE contrast showing similar phase assemblage as the fine-grained composite with niobium particles containing niobium oxide impurity within a fine-grained alumina matrix.

**Figure 11 materials-14-06453-f011:**
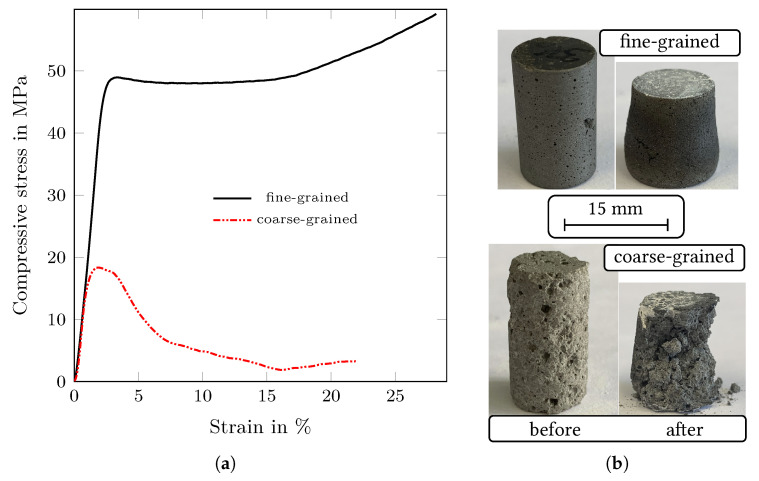
(**a**) Results of compression tests at a temperature of 1300 °C performed on the fine-grained and coarse-grained composites and (**b**) photographs of the corresponding specimens before and after the tests.

**Table 1 materials-14-06453-t001:** Determined particle sizes of the used raw materials and their chemistry.

Material	Particle Size in µm				Purity	Main Impurities
	$d_{10}$	$d_{50}$	$d_{90}$	$d_{\max}$		
niobium				75	99.95 wt.% Nb	O, C, Ta
Martoxid	0.13	0.63	3.57		99.8 wt.% Al${}_{2}$O${}_{3}$	Na, Si, Ca, Fe
CT9FG	1.96	5.53	20.63		99.5 wt.% Al${}_{2}$O${}_{3}$	Na, Fe, Si
CL370	0.17	0.54	5.77		99.7 wt.% Al${}_{2}$O${}_{3}$	Na, Si, Fe, Ca
Alphabond300		4-8		30	88 wt.% Al${}_{2}$O${}_{3}$ (min)	Ca, Na, Si

**Table 2 materials-14-06453-t002:** Castable recipe for the fine-grained composite.

Material	mass in g	Amount	
		in mass%	in vol.%
niobium	1644.9	79.95	64.31
Martoxid (alumina)	395.1	19.21	33.60
Alphabond300 (binder)	12.2	0.59	1.47
ADS-1 (dispersant)	5.2	0.25	0.62
water	112.9	5.20	27.0

**Table 3 materials-14-06453-t003:** Characteristic particle sizes of three synthesised composite aggregate classes measureable by laser granulometry.

Aggregate	Particle Size in µm		
**Class**	$d_{10}$	$d_{50}$	$d_{90}$
0–45 µm	0.2	11.7	50.4
45–500 µm	1.5	83.6	324.6
500–1000 µm	199.7	461.2	701.4

**Table 4 materials-14-06453-t004:** Determined values of envelope density $\rho_{b}$ and open porosity ${}_{a}$ according to DIN EN 993-1:2019-03, the skeleton density $\rho_{S}$ and porosity ε according to DIN ISO 15901-1:2019-03, and the true density $\rho_{F}$ according to DIN 66137-2:2019-03 of the synthesised aggregates as well as the sintered fine-grained and coarse-grained composites. If applicable, values are given as mean value ± standard deviation (number of samples n=4).

	Archimedes		MIP			He	XRD
	$\rho_{b}$	$\pi_{a}$	$\rho_{S}$	ε	$\rho_{b}$	$\rho_{F}$	ρ
	in g/$cm^{3}$	in %	in g/$cm^{3}$	in %	in g/$cm^{3}$	in g/$cm^{3}$	in g/$cm^{3}$
fine-grained composite	5.66±0.05	18.0±0.7	6.73	16.48	5.78		
**aggregate fraction**							
0–45 µm							5.899
45–500 µm						7.068	6.886
500–1000 µm						6.917	6.641
1000–3150 µm			6.88	15.83	5.79	6.918	6.661
coarse-grained composite	4.57±0.04	30.4±0.3	6.49	25.55	5.17		

**Table 5 materials-14-06453-t005:** Results of the Rietveld refinement of the four aggregate fractions 0–45 µm, 45–500 µm, 500–1000 µm and 1000–3150 µm. The overall fit quality of the Rietveld refinement can be judged with the presented weighted profile R-values $R_{\mathrm{wp}}$.

Aggregate Fraction	Phase (Amount in mass/vol.%)			Remarks
	Lattice Parameter in Å			
0–45 µm	**Nb-1 (28.8/20.0)**	**-Al${}_{2}$O${}_{3}$ (38.8/57.3)**	**-Nb${}_{2}$C (2.1/1.5)**	$R_{\mathrm{wp}}=7.22$
	a=3.31178	a=4.75892	a=5.36096	
		c=12.99214	c=4.96013	39.9 vol.% Nb
	**Nb-2 (28.8/19.9)**		**NbO (1.6/1.3)**	
	a=3.31045		a=4.21061	
45–500 µm	**Nb-1 (33.1/26.9)**	**-Al${}_{2}$O${}_{3}$ (20.3/35.0)**	**-Nb${}_{2}$C (2.6/2.2)**	$R_{\mathrm{wp}}=7.84$
	a=3.31575	a=4.75862	a=5.36064	
		c=12.9912	c=4.9614	60.3 vol.% Nb
	**Nb-2 (41.4/33.4)**		**NbO (2.6/2.4)**	
	a=3.30833		a=4.2103	
500–1000 µm	**Nb-1 (28.3/22.1)**	**-Al${}_{2}$O${}_{3}$ (24.7/40.9)**	**-Nb${}_{2}$C (2.5/2.1)**	$R_{\mathrm{wp}}=7.00$
	a=3.31557	a=4.75885	a=5.36271	
		c=12.99186	c=4.96004	55.0 vol.% Nb
	**Nb-2 (42.3/32.9)**		**NbO (2.2/2.0)**	
	a=3.30913		a=4.21104	
1000–3150 µm	**Nb-1 (37.6/29.5)**	**-Al${}_{2}$O${}_{3}$ (24.1/40.2)**	**-Nb${}_{2}$C (2.4/2.0)**	$R_{\mathrm{wp}}=7.49$
	a=3.31377	a=4.75866	a=5.36314	
		c=12.99121	c=4.95887	55.8 vol.% Nb
	**Nb-2 (33.7/26.3)**		**NbO (2.2/2.0)**	
	a=3.30858		a=4.21097	

**Table 6 materials-14-06453-t006:** Coarse-grained castable recipe.

Material	Density	mass	Amount	
	in g/$cm^{3}$	in g	in mass-%	in vol.%
0–45 µm	5.79	89.98	5.18	4.92
45–500 µm	5.79	533.02	30.69	29.12
500–1000 µm	5.79	447.5	25.77	24.45
1000–3150 µm	5.79	514.97	29.65	28.14
CL370 (alumina)	4.013	110.0	6.33	8.67
Alphabond300 (binder)	2.781	41.38	2.38	4.71
water	1	120	6.46	27.52

**Table 7 materials-14-06453-t007:** Determined values of the elastic constants Young’s modulus *E*, shear modulus *G* and Poisson’s ratio ν and the four-point bending strength σ of the sintered fine-grained and coarse-grained composites given as mean value ± standard deviation.

	Elastic Constants			MOR
	*E* in GPa	*G* in GPa	ν	σ in MPa
fine-grained	97.8±5.0	38.0±2.1	0.287±0.014	52.0±7.0
coarse-grained	46.0±1.8	24.5±2.2	0.185±0.021	11.3±0.4

## Data Availability

The data presented in this study are available on request from the corresponding author.

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
