# Peer review of "Synthesis of Niobium-Alumina Composite Aggregates and Their Application in Coarse-Grained Refractory Ceramic-Metal Castables"

_materials, 2021, doi:10.3390/ma14216453_

Round 1

Reviewer 1 Report

Authors submitted well-written article with generally high scientific impact and very strong technological part, but some minor issues must be improved:

  1. First of all, all tables and theirs captions are unreadable and placed in wrong places, what makes article difficult to read.
  2. Introduction part might be slightly improved, considering high-temperature applications of such composites.
  3. In materials and methods section, paragraph 2.1, lines 78-79 - the anode material used for X-ray diffraction should be added. Also, what scanning step was used? What exposition time per one step? Those are crucial information in XRD analysis and should be placed in the main text.
  4. In my opinion, Figure 4 is redundant - it do not shows significant informations.

Very strong part of this article are Rietveld refinement studies, which shows presence of two niobium phases. Authors suggestions about nature of this phenomena (impurities and/or residual stress) are right and should be strongly exposed in article. Moreover, authors should also consider performing TEM with SAED studies to confirm impurities or point defects in niobium.

Author Response

Please find our comments in the attached PDF file.

Reviewer 2 Report

General comments:

The manuscript is worthwhile and interesting for readers and it covers many issues related to Materials Design. Indeed, the experimental results on the room and high temperature mechanical properties, thermophysical properties such as the density and electrical conductivity as well as microstructural characterization together with an appropriate theoretical support are described in detail and can be used as guidelines for Materials Design. Speaking about the applications of refractory ceramics in metallurgy, either as containers for liquid metallic materials or as solids to be brazed it is necessary to write few words more on the wetting behaviour of metal/ceramic systems. Accordingly, in the Introduction, I suggest to insert the following:

“The wetting behaviour is depending on several properties, e.g. porosity and surface tension. “

Replace by

“The wetting behaviour of a liquid metal/ceramic system depends on the bonding characteristics of both system components as well as on the magnitude of interactive forces at the interface. It is characterised by the contact angle  of the liquid drop formed on the solid substrate indicating the chemical affinity between the two materials and by the work of adhesion  as a measure of interface adhesion strength [Novakovic R., Ricci E., Muolo M.L., Giuranno D., Passerone A. On the application of modelling to study the surface and interfacial phenomena in liquid alloy-ceramic substrate systems, Intermetallics 11(11-12) (2003) 1301-1311.]. Speaking about industrial applications of refractory ceramics, either as containers for liquid metallic materials or as solids to be brazed forming joints, it is necessary to know the interfacial properties (surface / interfacial tension and surface segregation) of melts and characteristics of solids (surface roughness, impurities, porosity) including the operating conditions such as temperature, time, gas composition and pressure [Eustathopoulos N, Nicholas M, Drevet B. Wettability at high temperatures. Pergamon materials series: vol 3. Oxford, UK: Pergamon; 1999.; Giuranno D., Bruzda G., Polkowska A., Nowak R., Polkowski W., Kudyba A., Sobczak N., Mocellin F., Novakovic R., Design of refractory SiC/ZrSi2 composites: Wettability and spreading behavior of liquid Si-10Zr alloy in contact with SiC at high temperatures, J. Eur. Ceram. Soc. 40(4) (2020) 953-960.]. It is important to mention that the surface energy can be influenced by applying a voltage to ground the refractory surface [14].”

“The latter can be influenced by applying a voltage to 33 ground the refractory surface [14]. “

Replace by

“It is important to mention that the surface energy can be influenced by applying a voltage to ground the refractory surface [14].”

Remark: Concerning the scientific terminology, usually the surface enegy of liquids is called the surface tension, while the term “the surface energy” is used for solids.

After Minor revisions, the Manuscript can be accepted for publications in Materials.

Author Response

Please find our answers in the attached PDF file.

Reviewer 3 Report

In the article is analysing niobium-alumina aggregates and possibilities to use them in fine-grained and coarse-grained castables. XRD, SEM, MIP analysis and many mechanical properties were tested.

Remarks:

Introduction must be improved by adding more information about in the article analysed castable properties: porosity, shrinkage, Young's modulus, etc. What influence they have to ceramic-metal castable quality and what requirements are for example for shrinkage, etc. What results were presented of other authors in similar area?

It was difficult to read article, because of many layout mistakes, unclear marks (Fig. 6, x-axis, Eq.1; Table 5, raw 311, etc.). Tables and Figures arranged away from the text.

From view of Figure 11b, it is seen, that coarse-grained sample was poorly formed (uneven surface and some huge pores). Don’t you think such specimens should be considered defective?

It isn't clear for what samples the electrical conductivity was tested and why only for 2 units?

In the Conclusions it is written, that created composites could be used for refractory linings. How about quite high shrinkage 4.3% of fine-grained composite and very low compressive strength of coarse-grained composite?

Author Response

Please find our replies in the attached PDF file.

Round 2

Reviewer 3 Report

The quality of the article was improved, but some layout errors remain (7p, 12p, 13p, 15p - Figure 8 is presented not in the same part of results).

Author Response

(The authors gave the same response as above.)
